# Efficient Similarity-Preserving Unsupervised Learning using Modular Sparse Distributed Codes and Novelty-Contingent Noise

## Abstract

There is increasing realization in neuroscience that information is represented in the brain, e.g., neocortex, hippocampus, in the form sparse distributed codes (SDCs), a kind of cell assembly. Two essential questions are: a) how are such codes formed on the basis of single trials, and how is similarity preserved during learning, i.e., how do more similar inputs get mapped to more similar SDCs. I describe a novel Modular Sparse Distributed Code (MSDC) that provides simple, neurally plausible answers to both questions. An MSDC coding field (CF) consists of $Q$ WTA competitive modules (CMs), each comprised of $K$ binary units (analogs of principal cells). The modular nature of the CF makes possible a single-trial, unsupervised learning algorithm that approximately preserves similarity and crucially, runs in fixed time, i.e., the number of steps needed to store an item remains constant as the number of stored items grows. Further, once items are stored as MSDCs in superposition and such that their intersection structure reflects input similarity, both fixed time best-match retrieval and fixed time belief update (updating the probabilities of all stored items) also become possible. The algorithm's core principle is simply to add noise into the process of choosing a code, i.e., choosing a winner in each CM, which is proportional to the novelty of the input. This causes the expected intersection of the code for an input, X, with the code of each previously stored input, Y, to be proportional to the similarity of X and Y. Results demonstrating these capabilities for spatial patterns are given in the appendix.

## 1 Introduction

Perhaps the simplest statement of the fundamental question of neuroscience is: how is information represented and processed in the brain, or what is the neural code? For most of the history of neuroscience, thinking about this question has been dominated by the "Neuron Doctrine" that says that the individual (principal) neuron is the atomic functional unit of meaning, e.g., that individual V1 simple cells represent edges of specific orientation and spatial frequency. This is partially due to the extreme difficulty of observing the simultaenous, ms-scale dynamics of all neurons in large populations, e.g., all principal cells in the L2 volume of a cortical macrocolumn. However, with improving experimental methods, e.g., larger electrode arrays, calcium imaging [26], there is increasing evidence that the "Cell Assembly" (CA) [9], a **set** of co-active neurons, is the atomic functional unit of representation and thus of cognition [29, 11]. If so, we have at least two key questions. First, how might a CA be assigned to represent an input based on a single trial, as occurs in the formation of an episodic memory? Second, how might similarity relations in the input space be preserved in CA space, as is necessary in order to explain similarity-based responding / generalization?

Submitted to 2nd Workshop on Shared Visual Representations in Human and Machine Intelligence (SVRHM).

I describe a novel CA concept, *Modular Sparse Distributed Coding* (MSDC), which provides simple, neurally plausible, answers to both questions. In particular, MSDC admits a single-trial, unsupervised learning method (algorithm) which approximately preserves similarity—specifically, maps more similar inputs to more highly intersecting MSDCs—and crucially, runs in **fixed time**. "Fixed time" means that the number of steps needed to store (learn) a new item remains **constant** as the number of items stored in an MSDC coding field (CF) increases. Further, since the MSDCs of all items are stored in superposition and such that their intersection structure reflects the input space's similarity structure, best-match (nearest-neighbor) retrieval and in fact, updating of the explicit probabilities of **all** stored items (i.e., "belief update" [17]), are also both fixed time operations.

There are three essential keys to the learning algorithm. 1) The CF has a modular structure: an MSDC CF consists of $Q$ WTA *Competitive Modules* (CMs), each comprised of $K$ binary units (as in Fig. 1). Thus, all codes stored in the CF or that ever become active in the CF are of size $Q$, one winner per CM. This modular CF structure distinguishes MSDC from numerous prior, "flat CF" sparse distributed representation (SDR) models, e.g., [28, 12, 16, 19]. 2) The modular organization admits an extremely efficient way to compute the *familiarity* ($G$, defined shortly), a generalized similarity measure that is sensitive not just to pairwise, but to all higher-order, similarities present in the inputs, without requiring explicit comparison of a new input to stored inputs. 3) A novel, **normative** use of noise (randomness) in the learning process, i.e., in choosing winners in the CMs. Specifically, an amount of noise inversely proportional to $G$ (directly proportional to novelty) is injected into the process of choosing winners in the $Q$ CMs. Broadly: a) to the extent an input is novel, it will be assigned to a code having low average intersection (high Hamming distance) with the previously stored codes, which tends to increase storage capacity; and b) to the extent it is familiar, it will be assigned to a code having higher intersection with the codes of similar previously stored inputs, which embeds the similarity structure over the inputs. The tradeoff between capacity maximization and embedding statistical structure is an area of active research [4, 14].

In this paper, I describe the MSDC coding format (Fig. 1), semi-quantitatively describe how the learning algorithm works (Figs. 2 and 3), then formally state a simple instance of the algorithm (Fig. 4), which shows that it runs in fixed time. The appendix includes results of simulations demonstrating the approximate preservation of similarity for the case of spatial inputs, and implicitly, fixed-time best-match retrieval and fixed-time belief update. This algorithm and model has been generalized to the spatiotemporal pattern (sequence) case [20, 23]: results for that case can be found in [22].

## 2 Modular Sparse Distributed Codes

Fig. 1b shows a simple model instance with an 8x8 binary pixel input, or receptive field (RF), e.g., from a small patch of lateral geniculate nucleus, which is fully (all-to-all) connected to an MSDC CF (black hexagon) via a binary weight matrix (blue lines). The CF is a set of $Q$=7 WTA *Competitive Modules* (CMs) (red dashed ellipses), each comprised of $K$=7 binary units. All weights are initially zero. Fig. 1a shows an alternate, linear view of the CF (which is used for clarity in later figures). Fig. 1c shows an input pattern, A, seven active pixels approximating an oriented edge feature, a code, $\phi(A)$, that has been activated to represent A, and the 49 binary weights that would be increased from 0 to 1 to form the learned association (mapping) from A to $\phi(A)$. Note: there are $K^Q$ possible codes.

Together, Figs. 2 and 3 describe a single-trial, fixed-time, unsupervised learning algorithm, made possible by MSDC, which approximately preserves similarity. A simple version of the algorithm, called the *code selection algorithm* (CSA), is then formally stated in Fig. 4. Fig. 2a shows the four inputs, A-D, that we will use to explain the principle for preserving similarity from an input space to the code space. Fig 2b shows the details of the learning trial for A. The model (a different instance than the one in Fig. 1), with A presenting as input, is shown at the bottom. The CF (gray hexagon) has $Q$=5 CMs, each with $K$=3 binary units. Since this is the first input, all weights are zero (gray lines). Thus, the bottom-up signals arriving from the five active input units yield raw input summation ($u$) of zero for the 15 CF units ($u$ charts). Note that we assume that all inputs have the same number, $S$=5, of active units. Thus, we can convert the raw $u$ values to normalized $U$ values in [0,1] by dividing by $S$ ($U$ charts). The final step is to convert the $U$ distribution in each CM into a probability distribution ($\rho$) from which a winner will be chosen. In this case, it is hopefully intuitive that the uniform $U$ distributions should be converted into uniform $\rho$ distributions ($\rho$ charts). Thus, the code chosen, $\phi(A)$, is completely random. Nevertheless, once $\phi(A)$ is chosen, the mapping from A to $\phi(A)$ is embedded at full strength, i.e., the 25 weights from the active inputs to the $Q$=5 winners are increased

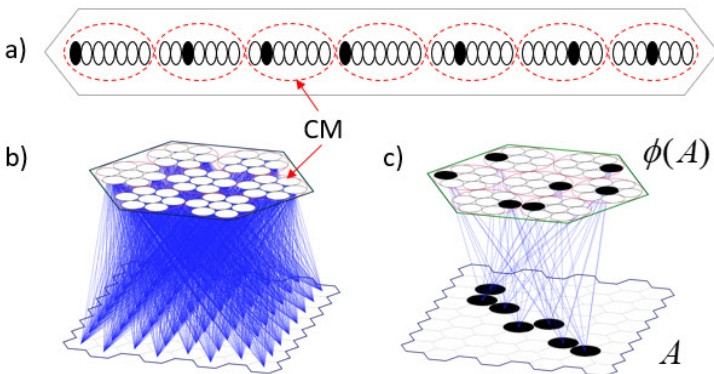

Figure 1: (a) Linear view of an MSDC coding field (CF) comprised of $Q$=7 WTA *competitive modules* (CMs) (red dashed boxes), each comprised of $K$=7 binary units. (b) A small model instance with an 8x8 binary pixel input, i.e., receptive field (RF), fully connected to the CF (black hexagon). (c) Example of learned association from an input, A, to its code, $\phi(A)$.

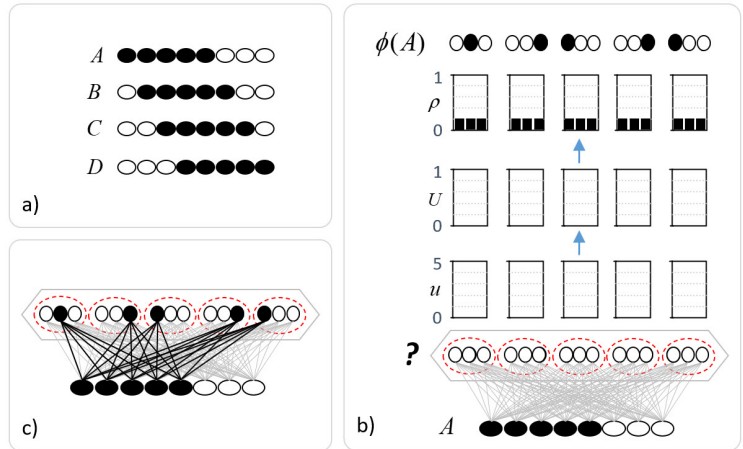

Figure 2: (a) Four sample inputs where B-D have decreasing similarity with A. (b) The learning trial for A. All weights are initially 0 (gray), thus $u=U=0$ for all units, which causes (see algorithm in Fig. 4) the win probability ($\rho$) of all units to be equal, i.e., uniform distribution, in each CM, thus a maximally random choice of winners as A's code, $\phi(A)$ (black units). (c) The 25 increased (from 0 to 1) weights (black lines) that constitute the mapping from A to $\phi(A)$.

from 0 to 1 (black lines in Fig. 2c). Thus, a strong memory trace can be immediately formed via the simultaneous increase of numerous weak (in an absolute sense) thalamocortical synapses, consistent with [2].

Input A having been stored (Fig. 2), Fig. 3 considers four hypothetical next inputs to illustrate the similarity preservation mechanism. Fig. 3a shows what happens if A is presented again and Figs. 3b-d show what happens for three inputs B-D progressively less similar to A. If A presents again, then due to the learning that occurred on A's learning trial, the five units that won (by chance) in that learning trial will have $u=5$ and thus $U=1$. All other units will have $u=U=0$. Again, it is hopefully intuitive in this case, that the $U$ distributions should be converted into extremely peaked $\rho$ distributions favoring the winners for $\phi(A)$. That is, in this case, which is in fact a retrieval trial, we want the model to be extremely likely to reactivate $\phi(A)$. Fig. 3a shows such highly peaked $\rho$ distributions and a statistically plausible draw where the same winner as in the learning trial is chosen in all $Q$=5 CMs. Fig. 3b shows the case of presenting an input B that is very similar to A (4 out of 5 features in common, red indicates non-intersecting input unit). Due to the prior learning, this leads to $u=4$ and $U=0.8$ for the five units of $\phi(A)$ and $u=U=0$ for all other units. In this case, we would like the model to pick a code, $\phi(B)$, for B that has high, but not total, intersection with

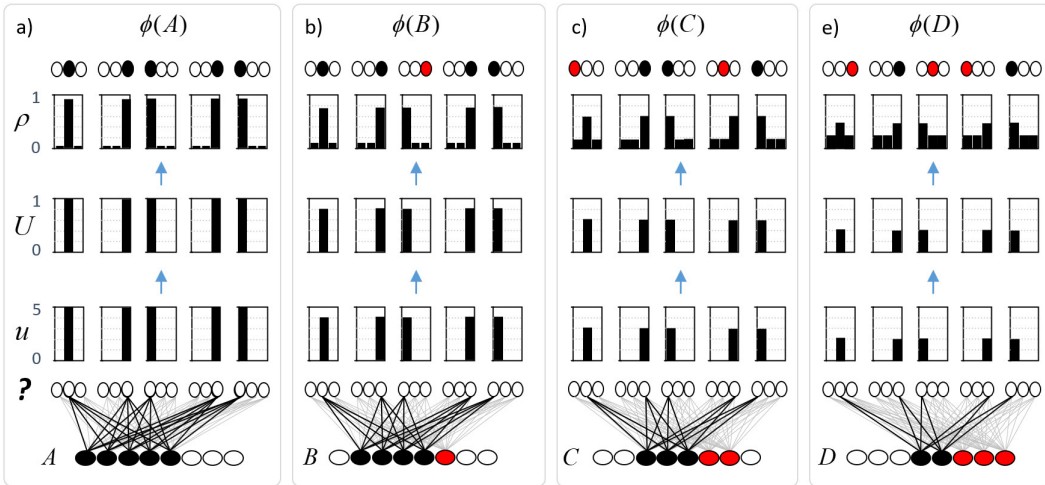

Figure 3: a) Illustration of the principle by which similarity is approximately preserved. Given that codes are MSDCs, all of size $Q$, all that needs to be done in order to ensure approximate similarity preservation is to make the probability distributions in the CMs increasingly noisy (flatter) in proportion to the novelty of the input.

$\phi(A)$. Clearly, we can achieve this by converting the $U$ distributions into slightly flatter, i.e., slightly noisier, $\rho$ distributions, than those in Fig. 3a. Fig. 3b shows slightly flatter $\rho$ distributions and a statistically plausible outcome where the most-favored unit in each CM (i.e., the winner for A) wins in four of the five CMs (red unit is not in intersection with $\phi(A)$). Figs 3c and 3d then just complete the explanation by showing two progressively less similar (to A) inputs, which lead to progressively flatter $\rho$ distributions and ultimately codes with lower intersections with $\phi(A)$. The $u$, $U$, and $\mu$ distributions are identically shaped across all CMs in each panel of Fig. 3 because each assumes that only one input, A, has been stored. As a succession of inputs are stored, the distributions will begin to differ across the CMs, due to the history of probabilistic choices (as can be seen in the appendix).

Having described the similarity preservation principle, i.e., adding noise proportional to input novelty into the code selection process, Fig. 4 formally states a simple version of the learning algorithm. Steps 1 and 2 have already been explained. Steps 3 and 4 together specify the computation of the *familiarity*, $G$, of an input, which is used to control the amount of noise added. $G$ is a generalized similarity measure and thus an inverse novelty measure. Step 3 computes the maximum $U$ value in each CM and Step 4 computes their average, $G$, over the $Q$ CMs. Steps 5 and 6 specify the nonlinear, specifically, sigmoidal, transform that will be applied from the $U$ values to relative probabilities of winning (within each CM) ($\mu$), which are then normalized to total probabilities ($\rho$). The main idea is as follows. If $G$ is close to 1, indicating the input, X, is highly similar to at least one stored input, Y, then we want to cause the units of $\phi(Y)$ to be highly favored to win again in their respective CMs. Thus, we put the $U$ values through a nonlinear transform that amplifies the differences between high and low $U$ values. On the other hand, if $G$ is near 0, indicating X is not similiar to any previously stored input, then we want to diminish the differences between high and low $U$ values, i.e., squash them together. Thus, we set the numerator, $\eta$, in Equation 6, to low value, which flattens the resulting $\rho$ distribution. When $G=0$, $\eta = 0$, and all units in the CM are equally likely to win. In work thus far, the $U$-to-$\mu$ transform (Step 6) has been modeled as sigmoidal. The motivation is that this will better model the phenomenon of categorical perception. However, a wider range of functions, including purely linear, would yield similarity preservation and should be investigated in future research.

Crucially, the learning algorithm has a **fixed** number of steps. That is, it iterates only over quantities that are fixed for the life of the model, i.e., the units and the weights, with only a single iteration occurring in any of the steps that involve iteration. In particular, there is no explicit iteration over stored inputs. This is an associative memory model, in a similar spirit to those of [28, 12], but with the added simple mechanism for statistically ensuring more similar inputs are assigned to more highly intersecting MSDCs.

| | Equation | Short Description |
|---|---|---|
| 1 | $u_i = \sum_{j \in RF} x_j w_{ji}$ | Compute weighted sum, $u_i$, of inputs $j$ in RF to unit $i$. $x_j \in \{0,1\}, w_{ji} \in \{0,1\}$ |
| 2 | $U_i = u_i / S$ | Normalize $u_i$. $S$ is the number of active units in an input pattern in the RF, which is assumed constant. |
| 3 | $\hat{U}_q = \max_{i \in C_q} U_i$ | Find max $U$, $\hat{U}_q$, in each CM, $C_q$. |
| 4 | $G \equiv \sum_{q=1}^{Q} \hat{U}_q / Q$ | Compute the *familiarity*, $G$, of the input as the average $\hat{U}_q$ value over the $Q$ CMs. $G$ is a generalized similarity measure, sensitive to pairwise and all higher-order similarities. |
| 5 | $\eta = G \times K \times \chi$ | Determine expansivity, $\eta$, of the activation function, which is modeled as a sigmoid transform (Eq. 6) from a unit's $U$ value to its relative (within its own CM) probability of winning, $\mu$. Factors $K$ (# of units in a CM) and $\chi$ (e.g., $\chi = 100$) influence probability of correct retrieval and stringency of the similarity criterion. |
| 6 | $\mu_i = \dfrac{\eta}{1 + e^{-\lambda(U_i - \phi)}} + 1$ | Transform each unit's $U$ value to unnormalized win probability, $\mu_i$. Note: the sigmoid collapses to a constant function when $\eta = 0$ (i.e., when $G = 0$). Parameters $\lambda$ and $\phi$ influence stringency of the similarity criterion, but in experiments to date, are not varied during model lifetime. |
| 7 | $\rho_i = \dfrac{\mu_i}{\sum_{k \in CM} \mu_k}$ | In each CM, normalize the relative probabilities ($\mu_i$) to the final probabilities ($\rho_i$) of winning. |
| 8 | Select final winner in each CM according to $\rho$-distribution in that CM (softmax). | |

Figure 4: Simple version of the learning algorithm sketched in Figs. 2 and 3.

## 3 Discussion

The work described herein has several novel components: 1) the modularity of sparse coding field; 2) the efficient means of computing familiarity ($G$); and 3) the normative use of noise to efficiently achieve approximate similarity preservation. Regarding (1), there is substantial evidence for the existence of mesoscale, i.e., macrocolumnar, coding fields in cortex, but we are as yet, a long way from definitively observing the formation (during learning) and reactivation/deactivation (during cognition/inference) of cell assemblies in such coding fields. Given that the model does learning and best-match retrieval, it can be viewed as accomplishing a form of locality sensitive hashing (LSH) [10], in fact, adaptive LSH (reviewed in [27]). Interestingly, recent work has proposed that the fly olfactory system performs a form of LSH [5, 6], and in fact, includes a novelty (i.e., inverse familiarity) computation, putatively performed by a mushroom body output neuron, that is quite similar to our model's $G$ computation. However, the Dasgupta et al model is not adaptive and thus, does not use novelty to influence the learning process. Finally, I emphasize the importance of the normative view of noise in our model. There has been much discussion of the nature, causes, and uses, of correlations and noise in cortical activity; see ([3, 13, 25]) for reviews. Most investigations of neural correlation and noise, especially in the context of probabilistic population coding models [30, 18, 8], assume *a priori*: a) fundamentally noisy neurons, and b) tuning functions (TFs) of some general form, e.g., unimodal, bell-shaped, and then describe how noise/correlation affects the coding accuracy of populations of cells having such TFs ([1, 15, 7, 24]). Specifically, these treatments measure correlation in terms of either mean spiking rates ("signal correlation") or spikes themselves ("noise correlations"). However, as noted above, our model makes neither assumption. Rather, in our model, noise (randomness) is actively injected—implemented via the $G$-dependent modulation of the neuronal transfer function—during learning to achieve the goal of similarity preservation. Thus, the pattern of correlations amongst units (neurons) simply emerges as a side effect of cells being selected to participate in MSDCs. How such a familiarity-contingent noise functionality might be implemented neurally remains an open question. It is most likely subserved by one or more of the brain's neuromodulatory systems, e.g., NE, ACh, and some preliminary ideas were sketched in [21].

## Broader Impact

I do not believe broader impact statement is applicable to this work.

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

# 4 Appendix

In this appendix, I present results of a small-scale simulation demonstrating approximate similarity
preservation for spatial inputs. In these experiments, the model has a 12x12 binary pixel input level
(i.e., receptive field, RF) that is fully connected to the CF, which consists of $Q$=24 WTA competitive
modules (CMs), each comprised of $K$=8 binary units. Fig. 5a shows six inputs, $I_1$ to $I_6$, all with
the same number of active pixels, $S$=12, which have been previously stored in the model instance
depicted in Fig. 3b. For simplicity of exposition, these six inputs have zero pixel-wise overlap with
each other. The second row of Fig. 3a shows a novel test stimulus, $I_7$, also with S=12 active pixels,
which has been designed to have progressively smaller pixel overlaps with and its varying overlaps
(yellow pixels) from $I_1$ to $I_6$. Given that all inputs are constrained to have exactly 12 active pixels,
we can measure input similarity simply as size of pixel intersection divided by 12 (shown as decimals
under inputs), e.g., $sim(I_x, I_y) = |I_x \cap I_y|/12$.

Fig. 5b shows the code, $\phi(I_7)$, activated in response to $I_7$, which by construction is most similar to
$I_1$. Black coding cells are cells that also won for $I_1$, red indicates active cells that did not win for
$I_1$, and green indicates inactive cells that did win for $I_1$. The red and green cells in a given CM can
be viewed as a substitution errors. The intention of the red color for coding cells is that if this is a
retrieval trial in which the model is being asked to return the closest matching stored input, $I_1$, then
the red cells can be considered errors. Note however that these are sub-symbolic scale errors, not
errors at the scale of whole inputs (hypotheses, symbols), as whole inputs are collectively represented
by the entire SDR code (i.e., be an entire "cell assembly"). In this example appropriate threshold
settings in downstream/decoding units, would allow the model as a whole return the correct answer
given that 18 out of 24 cells of 's code, $\phi(I_1)$, are activated, similar to thresholding schemes in other
associative memory models (Marr 1969, Willshaw, Buneman et al. 1969). Note however that if this
was a learning trial, then the red cells would not be considered errors: this would simply be a new
code, $\phi(I_7)$, being assigned to represent a novel input, $I_7$, and in a way that respects similarity in the
input space.

Fig. 5d shows the main message of the figure, and of the paper. The active fractions of the codes,
$\phi(I_1)$ to $\phi(I_6)$, representing the six stored inputs, $I_1$ to $I_6$, are highly rank-correlated with the
pixel-wise similarities of these inputs to $I_7$. Thus, the blue bar in Fig. 5d represents the fact that
the code, $\phi(I_1)$, for the best matching stored input, $I_1$, has the highest active code fraction, 75% (18
out 24, the black cells in Fig. 5b) of the cells of $\phi(I_1)$ are active in $\phi(I_7)$. The cyan bar for the next
closest matching stored input, $I_2$, indicates that 12 out of 24 of the cells of $\phi(I_2)$ (code note shown)

are active in $\phi(I_7)$. In general, many of these 12 may be common to the 18 cells in $\{\phi(I_7) \cap \phi(I_1)\}$. And so on for the other stored hypotheses. The actual codes, $\phi(I_1)$ to $\phi(I_6)$, are not shown; only the intersection sizes with $\phi(I_7)$ matter and those are indicated along right margin of chart in Fig. 3d. We note that even the code for $I_6$, which has zero intersection with $I_7$ has two cells in common with $\phi(I_7)$. In general, the expected code intersection for the zero input intersection condition is not zero, but chance, since in that case, the winners are chosen from the uniform distribution in each CM: thus, the expected intersection in that case is just $Q/K$.

As noted earlier, we assume that the similarity of a stored input, $I_Y$, to the current input, $I_X$, can be taken as a measure of $I_X$'s probability/likelihood. And, since all codes are of size $Q$, we can divide code intersection size by $Q$, yielding a normalized likelihood, e.g., $L(I_1) = |\phi(I_1) \cap \phi(I_y)|/Q$, as suggested in Fig. 5d. We also assume that $I_1$ to $I_6$ each occurred exactly once during training and thus, that the prior over hypotheses is flat. In this case the posterior and likelihood are proportional to each other, thus, the likelihoods in Fig. 5d can also be viewed as unnormalized posterior probabilities of the hypotheses corresponding to the six stored codes.

We acknowledge that the likelihoods in Fig. 5d may seem high. After all, $I_7$ has less than half its pixels in common with $I_1$, etc. Given these particular input patters, is it really reasonable to consider $I_1$ to have such high likelihood? Bear in mind that our example assumes that the only experience this model has of the world are single instances of the six inputs shown. We assume no prior knowledge of any underlying statistical structure generating the inputs. Thus, it is really only the relative values that matter and we could pick other parameters, notably in CSA Steps 5 and 6 of the learning algorithm, which would result in a much less expansive sigmoid nonlinearity, which would result in lower expected intersections of $\phi(I_7)$ with the learned codes, and thus lower likelihoods. The main point is simply that the expected code intersections correlate with input similarity, and thus, with likelihood.

Fig. 5c shows the second key message: the likelihood-correlated pattern of activation levels of the codes (hypotheses) apparent in Fig. 5d is achieved via independent soft max choices in each of the $Q$ CMs. Fig. 5c shows, for all 196 units in the CF, the traces of the relevant variables used to determine $\phi(I_7)$. As for Fig. 3, the raw input summation from active pixels is indicated in the $u$ charts. Note that while all weights are effectively binary, "1" is represented with 127 and "0" with 0. Hence, the maximum $u$ value possible in any cell when $I_7$ is presented is 12x127=1524. The normalized input summations are given in the $U$ charts. As stated in Fig. 4, a cell's $U$ value represents the total local evidence that it should be activated. However, rather than simply picking the max $U$ cell in each CM as winner (i.e., hard max), which would amount to executing only steps 1-3 of the learning algorithm, the remaining CSA steps, 4-8, are executed, in which the $U$ distributions are transformed as described in Fig. 4 and winners are chosen via soft max in each CM. The final winner choices, chosen from the $\rho$ distributions are shown in the row of triangles just below CM indexes. Thus, an extremely cheap-to-compute (ie., Step 4) *global* function of the whole CF, $G$, is used to influence the local decision process in each CM. We repeat for emphasis that no part of the algorithm explicitly operates on, i.e., iterates over, stored hypotheses; indeed, there are no explicit (localist) representations of stored hypotheses on which to operate; all items are stored in sparse superposition.

Fig. 6 shows that different inputs yield different likelihood distributions that correlate approximately with similarity. Input $I_8$ (Fig. 6a) has highest intersection with $I_2$ and a different pattern of intersections with the other learned inputs as well (refer to Fig. 5a). Fig. 6c shows that the codes of the stored inputs become active in approximate proportion to their similarities with $I_8$, i.e., their likelihoods are simultaneously physically represented by the fractions of their codes which are active. The $G$ value in this case, 0.65, yields, via steps 5 and 6, the $U$-to-$\mu$ transform shown in Fig. 6b, which is applied in all CMs. Its range is [1,300] and given the particular $U$ distributions shown in Fig. 6d, the cell with the max $U$ in each CM ends up being greatly favored over other lower-$U$ cells. The red box shows the $U$ distribution for CM 9. The second row of the abscissa in Fig. 6b gives the within-CM indexes of the cells having the corresponding (red) values immediately above (shown for only four cells). Thus, cell 3 has $U$=0.74 which maps to approximately $\mu$ =250 whereas its closest competitors, cells 4 and 6 (gray bars in red box) have $U$=0.19 which maps to $\mu$ =1. Similar statistical conditions exist in most of the other CMs. However, in three of them, CMs 0, 10, and 14, there are two cells tied for max $U$. In two, CMs 10 and 14, the cell that is *not* contained in $I_2$'s code, $\phi(I_2)$, wins (red triangle and bars), and in CM 0, the cell that is in $\phi(I_2)$ does win (black triangle and bars). Overall, presentation of $I_8$ activates a code $\phi(I_8)$ that has 21 out of 24 cells in common with $\phi(I_2)$ manifesting the high likelihood estimate for $I_2$.

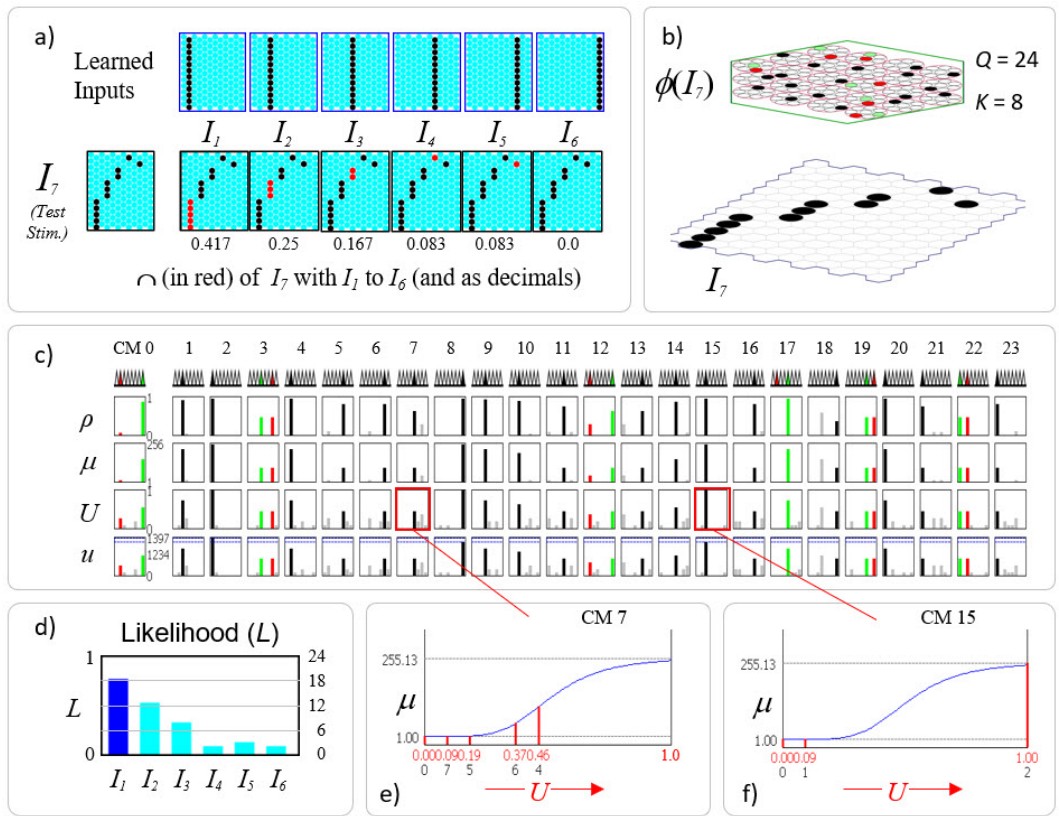

Figure 5: In response to an input, the codes for learned (stored) inputs, i.e., hypotheses, are activated with strength that is correlated with the similarity (pixel overlap) of the current input and the learned input. Test input $I_7$ is most similar to learned input $I_1$, shown by the intersections (red pixels) in panel a. Thus, the code with the largest fraction of active cells is $\phi(I_1)$ (18/24=75%) (blue bar in panel d). The other codes are active in rough proportion to the similarities of $I_7$ and their associated inputs (cyan bars). (c) Raw ($u$) and normalized ($U$) input summations to all cells in all CMs. The $U$ values are transformed to unnormalized win probabilities ($\mu$) in each CM via a sigmoid transform whose properties, e.g., max value of 255.13, depend on $G$ and other parameters. The $\mu$ values are normalized to true probabilities ($\rho$) and one winner is chosen in each CM (indicated in row of triangles: black: winner for $I_7$ that also won for $I_1$; red: winner for $I_7$ that did not win $I_1$: green: winner for $I_1$ that did not win for $I_7$. (e, f) Details for CMs, 7 and 15. Values in second row of $U$ axis are indexes of cells having the $U$ values above them. Some CMs have a single cell with much higher $U$ and ultimately $\rho$ value than the rest (e.g., CM 15), some others have two cells that are tied for the max (e.g., CMs 3, 19, 22).

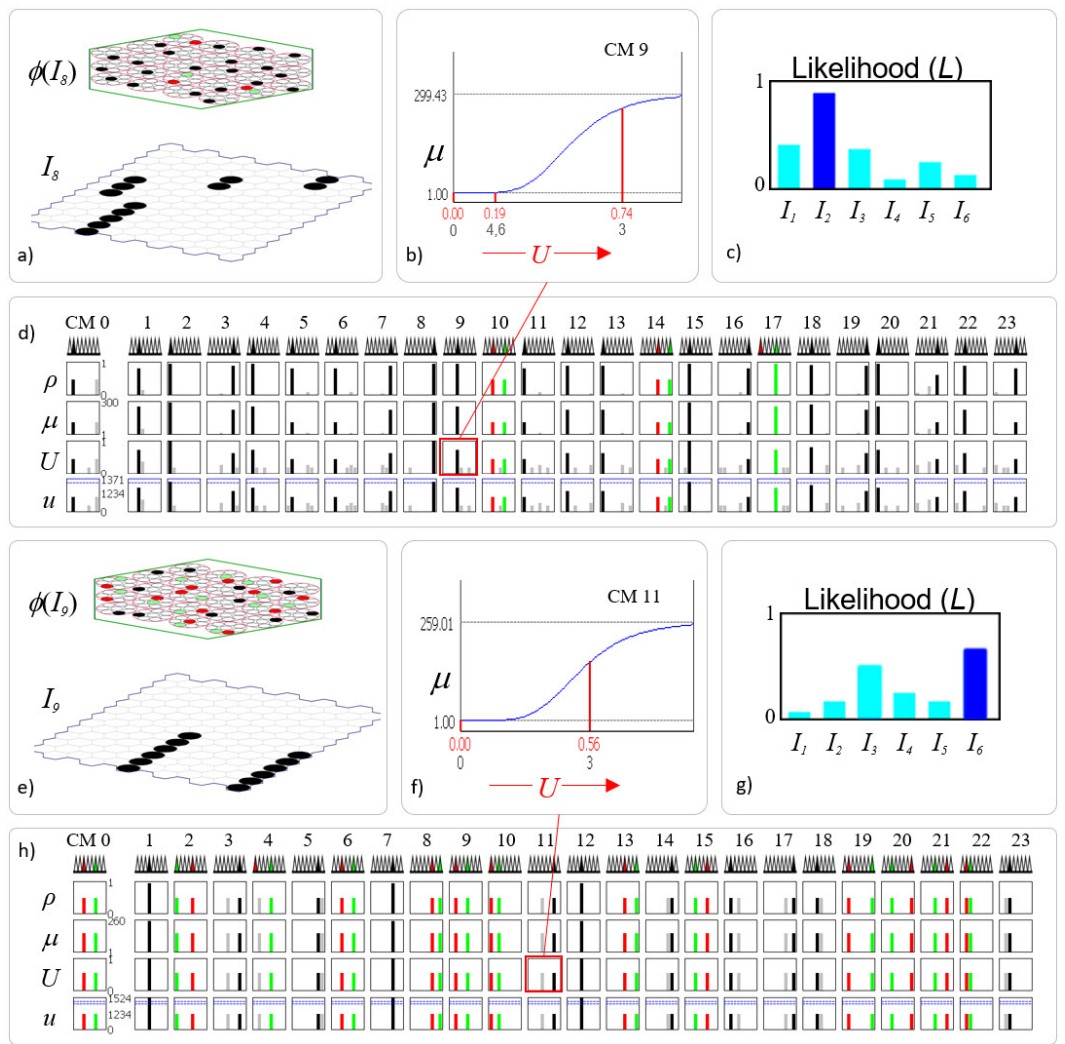

Figure 6: Details of presenting two further novel inputs, (panels a-d) and (panels e-h). In both cases, the resulting likelihood distributions correlate closely with the input overlap patterns. Panels b and f show details of one example CM (indicated by red boxes in panels d and h) for each input

.

To finish our demonstration of approximate similarity preservation, Fig. 6e shows presentation of another input, $I_9$, having half its pixels in common with $I_3$ and the other half with $I_6$. Fig. 6g shows that the codes for $I_3$ and $I_6$ have both become approximately equally (with some statistical variance) active and are both more active than any of the other codes. Thus, the model is representing that these two hypotheses (stored items) are the most likely and approximately equally likely. The exact bar heights fluctuate somewhat across repeated trials, e.g., sometimes $I_3$ has higher likelihood than $I_6$, but the general shape of the distribution is preserved. The fact that one of the two bars is blue, the other cyan, just reflects the approximate nature of the retrieval process. The remaining hypotheses' likelihoods also approximately correlate with their pixelwise intersections with $I_9$. The qualitative difference between presenting $I_8$ and $I_9$ is readily seen by comparing the $U$ rows of Fig. 6d and 6h and seeing that for the latter, a tied max $U$ condition exists in almost all the CMs, reflecting the equal similarity of $I_9$ with $I_3$ and $I_6$. In approximately half of these CMs, the cell that wins intersects with $\phi(I_3)$ and in the other half, the winner intersects with $\phi(I_6)$. In Fig. 6h, the three CMs in which there is a single black bar, CMs 1, 7, and 12, indicates that the codes, $\phi(I_3)$ and $\phi(I_6)$, intersect in those CMs.