# OpenReview forum: "Efficient Similarity-Preserving Unsupervised Learning using Modular Sparse Distributed Codes and Novelty-Contingent Noise"
_NeurIPS.cc/2020/Workshop/SVRHM — Submitted to SVRHM@NeurIPS_

### Official Review · AnonReviewer3 · 2020-10-23
**Interesting ideas but difficult to evaluate**

**Rating:** 4
**Confidence:** 2

**Review:**


There are some interesting ideas here, but their dense presentation makes the paper's contribution difficult to crystallize. For all the discussion of mechanisms and the proposed model, the paper is also lacking in quantitative results or other comparisons to make the paper's contribution understandable.

Some larger issues:

- neural inputs are not binary. Is the binary nature of this model required or only for computational / didactic purposes? How would the model perform on nonbinary inputs? Given that the demonstrations of the model and its operation (Appendix) all use simple toy examples, I must wonder whether some other factor(s) limit the model's applicability to more standard problem sets (e.g. at least MNIST)?

- Why is it crucial that a model of the neural code runs in fixed time? What evidence is there that learning in the brain runs in fixed time?

- The paper is dense in non-standard abbreviations and mathematical notation, making it difficult to follow the central argument.

- Is it neurally plausible that the noise in the process is proportional to novelty -- novelty is presumably computed *after* inputs have noise added?

- S (number of active units) assumed constant. Is this realistic (how could a cortical column know S? Doesn't this limit the kinds of "input stimulus" that can be processed?), and how does the performance of the learning algorithm depend on this assumption?

## Minor points

- grammatical problems (e.g. abstract, starting a list with "a)" but not following with a subsequent letter).

- a number of sweeping statements are not adequately supported by the cited literature:

"There is increasing realization in neuroscience that information is represented in the1brain, e.g., neocortex, hippocampus, in the form sparse distributed codes (SDCs), a kind of cell assembly."

-- is there? What evidence supports this statement?

"there is increasing evidence that the “Cell Assembly” (CA) [9], a set of co-active neurons, is the atomic functional unit of representation and thus of cognition [29, 11]"

Do these two citations (one on memory, one on networks) support the "increasing realization... that information is represented... in the form of sparse distributed codes"?

- Checking my understanding: it seems that the only way for familiarity G to be zero is if all u_i (weighted sums of inputs) are zero? (i.e. all inputs j or weights are zero)?

"I do not believe broader impact statement is applicable to this work."

- This is not good enough. The out-of-the-box thinking evident in the paper can also be applied here.

---

### Official Review · AnonReviewer2 · 2020-10-27
**a submission with some intriguing ideas that are not clearly stated and it is unclear whether they are significant**

**Rating:** 4
**Confidence:** 3

**Review:**

The paper entitled “Efficient Similarity-Preserving Unsupervised Learning using Modular Sparse Distributed Codes and Novelty-Contingent Noise ” described a coding scheme called “modular sparse distributed code (MSDC)”. This coding scheme consists a number of WTA models. It is claimed that the model can perform single-trial, unsupervised learning, and preserve the similarity structure in the input.
I am not an expert on the sparse modular code, but I am generally familiar with the neural coding and population coding literature. I have to say that this paper is not particularly well-written, and I am not sure if I fully understood the paper. It appears to me that the paper basically describes a template matching procedure with some normalization. It is unclear to me what extent the model (described in Fig 4) is novel. It is also unclear to me exactly what learning rule is being used (is it Hebbian? is it one-shot?).  The computational problem described should have both learning and inference. However, it is not clear how to distinguish learning and inference in the model. For example, when and how to add new memories?

Overall, while I think this manuscript contains several interesting ideas (in particular the idea of using noise to preserve the similarity) which may be nice to be discussed in the workshop, I also have some pretty serious concerns about it.
Clarity: I think the clarity could be much improved. The manuscript would benefit by improving the presentations, e.g., better connecting to the previous literature and clearly state what is novel and what is not.
Significance: Although the ideas are interesting,  in their current form, I don’t quite see how they could significantly advance our understanding of neural coding.

Below I list several questions/issues/suggestions as I read through the manuscripts.

* The paper emphasizes that the algorithm runs in fixed time. However, this is not motivated- it is unclear why this is important, and it is also unclear whether the brain in fact satisfies this property.

* The model ignores realistic neural noise, and it assumes neurons are binary. It is proposed that neural noise can be injected to preserves the similarity. I do find that this is a very interesting and intriguing idea, but naturally a question needs to be addressed is how this could be achieved when (at least part of ) the neural variability are due to intrinsically stochasticity(such as biophysical noise etc), thus could not directly controlled. Also, would it still work if neurons have continuous firing rate?

* Also what is the capacity of the code? It seems that this is not explicitly described (sorry if I missed it).  I am also under the impression that the capacity of this code does not exceed the number of modules.

* The WTA models are not directly implemented with neural dynamics via recurrent connections. Maybe it would be useful to consider adding recurrent connections in the model?

*How does this model compare to the Hopfield network? Maybe in fact it is an abstraction of what Hopfield network is doing  but without the dynamics?

 *it would be useful to add some quantifications on how well the similarity is in fact preserved.

*The model seems mostly like a memory model (e.g., an extremely simplified model of hippocampus). I am not sure how it connects to vision and the main thesis of this particular workshop. Maybe I am missing something.

---

### Official Review · AnonReviewer1 · 2020-10-27
**Review: Efﬁcient Similarity-Preserving Unsupervised Learning using Modular Sparse Distributed Codes and Novelty-Contingent Noise**

**Rating:** 4
**Confidence:** 2

**Review:**

This work describes a learning algorithm that is intended to be a neurally plausible implementation of the cell assembly concept. The presented model allows to process single trials and ensures that similar trials have similar representations. The similarity to previous trials is thereby assessed without relying on pairwise comparisons. Instead, intermediate representations are used, which allows a more efficient evaluation in fixed time. The estimated similarity determines the amount of noise that is added when the representation is computed. The noise directly influences the similarity of the representation of the current trial to the representations of the previously seen trials. The figures are very helpful to illustrate the proposed learning procedure. The appendix provides a small scale example that shows that the learned representations indeed preserve similarity.

- The presented ideas do not seem contemporary to me. I'm missing an overview of related work that describes how this learning algorithm compares to other unsupervised learning methods.
- While the model might be able to find a representation that reflects input similarity, it is not clear to me, why it should learn a meaningful representation.
- In general, I found the paper hard to read. The following aspects contributed to that:
 - In the abstract an enumeration is started with an "a)", but there is no "b)" following (line 3).
 - There are quite a few definitions and explanations of abbreviations in the abstract that reduce the readability. For example, why are the parameters describing the architecture (Q and K in line 7-9) explained in the abstract?
 - There are some weird sentences, such as "Input A having been stored (Fig. 2), Fig. 3 considers..." (line 94).
 - The variable $\mu$ used in line 112-115 was not introduced before usage.

---

### Decision · Program_Chairs · 2020-11-02

Reject